# AI Applied to Volatile Organic Compound (VOC) Profiles from Exhaled Breath Air for Early Detection of Lung Cancer

**DOI:** 10.3390/cancers16122200

**Published:** 2024-06-12

**Authors:** Manuel Vinhas, Pedro M. Leitão, Bernardo S. Raimundo, Nuno Gil, Pedro D. Vaz, Fernando Luis-Ferreira

**Affiliations:** 1Departamento de Engenharia Electrotécnica e de Computadores, Faculdade de Ciências e Tecnologia, Universidade Nova de Lisboa, 2829-516 Monte da Caparica, Portugal; mfp.vinhas@campus.fct.unl.pt; 2Unidade de Pulmão, Centro Clínico Champalimaud, Fundação Champalimaud, Av. Brasília, 1400-038 Lisbon, Portugal; pm.leitao@campus.fct.unl.pt (P.M.L.); bernardo.raimundo@research.fchampalimaud.org (B.S.R.); nuno.gil@fundacaochampalimaud.pt (N.G.)

**Keywords:** lung cancer, volatile organic compounds, artificial intelligence, early detection, machine learning

## Abstract

**Simple Summary:**

Lung cancer stands as a serious health challenge, prompting the exploration of innovative detection methods. Volatile organic compounds (VOCs) found in exhaled breath air are becoming a relevant opportunity for early cancer detection, including lung cancer, without invasive procedures or high costs. Unlike traditional approaches, which target specific compounds, this study analyzes overall compositional profiles, maximizing detection efficiency. The results highlight the potential of AI-driven techniques in revolutionizing early cancer detection for clinical use.

**Abstract:**

Volatile organic compounds (VOCs) are an increasingly meaningful method for the early detection of various types of cancers, including lung cancer, through non-invasive methods. Traditional cancer detection techniques such as biopsies, imaging, and blood tests, though effective, often involve invasive procedures or are costly, time consuming, and painful. Recent advancements in technology have led to the exploration of VOC detection as a promising non-invasive and comfortable alternative. VOCs are organic chemicals that have a high vapor pressure at room temperature, making them readily detectable in breath, urine, and skin. The present study leverages artificial intelligence (AI) and machine learning algorithms to enhance classification accuracy and efficiency in detecting lung cancer through VOC analysis collected from exhaled breath air. Unlike other studies that primarily focus on identifying specific compounds, this study takes an agnostic approach, maximizing detection efficiency over the identification of specific compounds focusing on the overall compositional profiles and their differences across groups of patients. The results reported hereby uphold the potential of AI-driven techniques in revolutionizing early cancer detection methodologies towards their implementation in a clinical setting.

## 1. Introduction

Lung cancer continues to pose a significant public health challenge in Europe, with high mortality rates associated with the disease [1]. According to recent statistics, lung cancer remains the leading cause of cancer-related deaths across Europe, accounting for approximately 244,000 fatalities predicted for 2024 [2,3]. Despite advancements in treatment strategies, the overall survival rate for lung cancer remains relatively low, primarily due to late-stage diagnoses.

The importance of early detection in combating lung cancer cannot be overstated. Recently published research underscored the critical role of early detection through screening programs, which could lead to the identification of lung cancer at more treatable stages [2]. Early-stage lung cancer offers better treatment options and significantly improved survival rates compared to advanced-stage disease. Nevertheless, detection is hampered by the lack of abundance of circulating tumor DNA (ctDNA) in early-stage disease [4].

However, challenges persist in current diagnostic techniques for lung cancer, particularly regarding invasiveness. Conventional diagnostic methods such as bronchoscopy and biopsy often entail invasive procedures that carry inherent risks and discomfort for patients [5]. Additionally, these invasive techniques may not always be suitable for patients with underlying health conditions or compromised overall health. Decisions about widely adopted screenings are limited by diagnostic accuracy and false positives, which present significant hurdles in lung cancer detection. The issue of false-positive results in lung cancer screening programs, leading to unnecessary follow-up procedures, patient anxiety, and increased healthcare costs, has been addressed in some studies [6]. Striking a balance between sensitivity and specificity in screening tests is crucial to minimize false positives while ensuring the timely detection of true lung cancer cases.

Addressing the persistently high mortality rates associated with lung cancer in Europe emphasizes the need for a comprehensive approach that prioritizes early detection, minimizes false positives, and enhances the accuracy of screening methods. Studies have underlined the importance of early detection in improving treatment success and reducing mortality rates, in which the benefits of low-dose computed tomographic screening in reducing lung cancer mortality are highlighted [7,8]. Additionally, efforts to pursue non-invasive diagnostic techniques are crucial in overcoming the limitations of invasive procedures and improving patient outcomes. Therefore, developing non-invasive biomarkers for the early detection of lung cancer can be a feasible option [9]. Moreover, addressing the issue of false positives is essential to prevent unnecessary interventions and reduce patient anxiety. The need to find a balance between sensitivity and specificity in screening tests to minimize false positives while ensuring the timely detection of true lung cancer cases must be the driving force [10]. It is therefore of utmost relevance to enhance the accuracy of screening methods capable of reliably detecting lung cancer at its earliest stages. Significant strides can be made in improving treatment success and ultimately reducing mortality rates. The current research endeavors to develop a non-invasive method using analysis of exhaled breath air for detecting cancer at an early stage [11,12,13,14]. This approach offers the advantage of enhancing adherence to screening techniques by minimizing discomfort and ensuring a high level of reliability, as elaborated in the following sections.

## 2. Experimental Methodology

### 2.1. Volunteer Recruitment and Characterization

The present study was carried out at Champalimaud Clinical Centre, Champalimaud Foundation in Lisbon, Portugal. The study was approved by the ethics committee of Champalimaud Foundation under registered codename VOX-PULMO and all procedures were under good medical practice. The population was recruited between July 2020 and December 2023 comprising the following two groups: (1) patients with diagnosed lung cancer, and (2) a control group of healthy subjects. Patients eligible for recruitment who fulfilled the eligibility criteria were referred by the clinical staff through multidisciplinary team meetings and recruited. The eligibility criteria at the time of enrollment were:For the lung cancer group: being eighteen (18) years old or older; (2) able to understand language spoken and/or written; (3) able to provide consent based on previous information; (4) with diagnosed disease of lung cancer independent of its histology; (5) without any other cancer disease at another primary location.For the control group: being eighteen (18) years old or older; (2) able to understand language spoken and/or written; (3) able to provide consent based on previous information; (4) without other infectious or inflammatory diseases; (5) without any oncologic disease.

Once recruited, patients were validated by the medical staff at the clinical site. The volunteers’ background data—demographic profiles, cigarette smoking history, staging, pathological findings—were collected during the recruitment appointment. The clinical cancer stage was based upon the American Joint Committee on Cancer (AJCC) TNM staging system, 8th edition [15].

### 2.2. Exhaled Breath Sampling and Chemical Analysis

Samples and data were collected before any type (local or systemic) of cancer treatment in the LC patient’s group. Breath samples were collected at Champalimaud Foundation Clinical Centre. Atmospheric air from sample collection rooms and from the sample collection device was also analyzed to investigate the effects of background VOCs on collected breath samples. The samples for background evaluation were sampled according to a standard literature procedure for this device [16,17]. Breath sampling was conducted into thermal desorption (TD) tubes (bio-monitoring, inert coated tubes, Markes International Ltd.) using a ReCIVA breath sample system (Owlstone Medical, Cambridge, UK), coupled to a clean air supply comprising a pump connected to an active charcoal scrubber (CASPER system; Owlstone Medical, Cambridge, UK). Collection of breath is conducted directly with a mask holding the TD tubes. Each volunteer provided a total volume of 2 L of full tidal exhaled breath that was transferred onto the TD tubes, at a flow rate of 500 mL/min. Total time of sample collection was usually ca. 4 min.

All samples were analyzed by gas chromatography coupled to field-asymmetric ion mobility spectrometry (GC-FAIMS) [18], using a commercially available instrument (Lonestar, Owlstone, Cambridge, UK). FAIMS is based on an oscillating electric field to separate different gas-phase ions based on their different mobility across an electrical field relating to size and mass. The resulting data are a pattern of a “chemical fingerprint” (the total chemical composition) of a given sample rather than the individual components.

## 3. Results

The main objective of this study was to classify exhaled breath air profiles collected from a set of individuals with and without lung cancer alongside their background data using artificial intelligence algorithms. These used the chemical data obtained from analysis on the GC-FAIMS spectrometer. Details will be set in the following sections.

### 3.1. Patient Characterization

The study recruited a total of 203 validated participants from both clinical sites, distributed as 77 within the lung cancer (LC) and 126 within the healthy control (HC) groups, respectively. All volunteers in the LC group had their disease confirmed by a clinical protocol. Those from the control group mandatorily could not have any known cancer until the time of sample collection. The background demographic data for the participants in this study by volunteer group are shown in Table 1.

According to Table 1, the population with LC showed a median age of 66 years (range, 41–86 years), being older than that the HC group, which showed a median age of 40 years (range, 20–78 years). The median age of the former group matches exactly that reported for patients with lung cancer in Portugal [19], whereas the median age of the latter group is close to that of the overall Portuguese population, according to the Portuguese National Statistics Office [20]. The data from Table 1 also made it possible to find that the proportion of current smokers was matched across both groups (ca. 31%). On the other hand, the fraction of never-smokers was 31.2% (similar to that of the current smokers in the LC cohort as opposed to more than one half (55.9%) in the HC group).

### 3.2. The Proposed Methodology

Exhaled volatile organic compounds (VOCs), arising from the body’s metabolic activity, mirror an organism’s condition at a given moment in time [21,22,23,24]. As such, the two studied cohorts had their exhaled breath profiles evaluated. The use of GC-FAIMS yielded a complex profile of volatile metabolites based on intensities for each volunteer that could be classified according to their status (control or LC cohorts) [25]. As shown in Figure 1, it is possible to see that the profiles from both healthy controls and lung cancer groups differ clearly.

The overall collected data from patients in the trial resulted in a set of 203 images each with 408 pixels width and 180 pixels height (Figure 1). The image’s color is made with 3 color channels (RGB) with 8 bits. The code specifically created for the study consists of an application developed for images resulting from spectroscopy, where all the architecture was implemented using Keras API from the Tensorflow library, including all the model compilation, fitting, and evaluation. The results in terms of plots and metrics were implemented with the scikit-learn library. The hardware specification used is detailed in Appendix A at the end of this article.

### 3.3. Dataset Structure

The objective of the classification task was to group the images into two different classes: Class 0, which included the negative samples for lung cancer volunteers (healthy controls), and Class 1, which had all the samples of patients that are positive for lung cancer (lung cancer patients).

The classification task consisted of the implementation of machine learning (ML) supervised algorithms [27]. For the start of the analysis, the results from the spectrometer were prepared to feed the engine, mostly by trimming the non-relevant information, thus resulting in two distinct datasets. Each dataset was used to train, validate, and test the model. In this study, each sample in the image datasets was labeled according to the corresponding lung cancer status, classified in a binary mode as 1 for cancer or 0 for healthy.

Table 2 presents the resulting datasets with a data split of approximately 25% of all cases on the test dataset and the remaining 75% on the training and validation datasets. For some algorithms, which will be referred to later, the training and validation datasets were merged, having, in this case, those exact percentages.

Considering all the images, the positive cases accounted for approximately 38% of all images, while the remaining 62% accounted for the negative cases of lung cancer. It was then ensured that the proportion of each dataset conserved this percentual relation.

With the datasets already defined, different pre-processing methods were applied. In the first option, the images are converted from the RGB color space to grayscale (see Figure 1 for examples). At the same time, the pixel values were normalized from the interval [0, 255] to [0, 1], which was globally preferred for several reasons, such as to avoid exploding gradients during training.

The second method used the image with the original 3 color channels, but this time the image histogram was equalized with contrast-limited adaptative histogram equalization (CLAHE) [26]. The objective of this method was to make some hidden (less bright) patterns of the image more evident. Besides this, the normalization was the same as the first method, so that a final image had pixel values in the interval [0, 1]. The plots depicted in Figure 2 illustrate the variations in graphics resulting from the application of each data preparation method.

A sample of a grayscale image with no histogram equalization is presented in Figure 2a, with all pixel intensities located close to 0, meaning that most of the pixels (frequency on the graph) had a value less than 50 (pixel intensity). Consequently, the image exhibited a darker appearance, as lower pixel values corresponded to darker tones.

Figure 2b shows the results from the contrast-limited adaptative histogram equalization (CLAHE) of the RGB image. It is possible to see that the peak appearing in Figure 2a has moved to the right, meaning that the image’s pixel values have increased, making the dark characteristic of the image less dark. Therefore, less visible patterns became lighter, clearer, and more visible.

### 3.4. Experimental Methodology

The research involved the application of various machine learning (ML) algorithms to construct classification models. A series of tests were conducted to evaluate the performance of each model. Throughout the training phase, accuracy and loss were recorded for each epoch, leading to a graphical representation of these results. Once assessing the models with the test dataset, an initial outcome emerged in the form of a confusion matrix, depicting the correlation between predicted labels and true labels. Following this, two tables were generated, containing essential metrics. One table presented the relationship between weighted and macro averages and metrics such as precision, recall, F1 score, and support. Similarly, the other table illustrated the association between class labels (0 or 1) using the same metrics, mirroring the structure of the first table.

From here, two more plots were obtained, the ROC (receiver operating characteristic) and the precision–recall curve. Both these plots resulted from the variation of the classification threshold to assess how the true positive rate (TPR) over the false positive rate (FPR) and how the precision (recall) varied for the ROC and precision–recall curves varied, respectively.

The original classification threshold was set to 0.5 to distinguish between the positive and the negative classes from which the confusion matrix was derived. For each tested model, a table presenting the accuracy loss AUC (area under the curve) and cross-validation results was created.

#### 3.4.1. GoogLeNet

The GoogLeNet Inception V1 deep neural network, composed of five main blocks, was used for the study [28]. For all tests, the layers, padding, strides, and the activation function were maintained with the following values: same padding, (2,2) and a non-linear activation function, rectified linear unit (ReLU) as any exception to this will be highlighted.

The first block had three 2D convolution layers (2DConvs), whereas the first layer had 64 filters with a dimension of 7 × 7 pixels followed by a Max pooling layer with a kernel of 3 × 3 [29]. After this layer, two more 2DConvs with 64 and 192 as the number of filters, with dimensions of 1 × 1 and 3 × 3, respectively. This block ended with a 2D Max pooling layer, with the same parameters as before, that served to reduce the spatial dimensions and increase the number of channels, preparing the extracted feature maps for the inception blocks.

The first couple of inception modules had the specified array with the number of filters for the inception module: (64, 96, 128, 16, 32, 32) and (128, 128, 192, 32, 96, 64), applied sequentially, followed by a 2D Max pooling layer with 3 × 3 kernel size.

The next block was composed of five inception modules with the following parameters for the number of filters: (64, 96, 128, 16, 32, 32) and (128, 128, 192, 32, 96, 64), applied sequentially, followed by a 2D Max pooling layer with a 3 × 3 kernel size. This block extracted a wide variety of features, which could be described as mid-level features.

The next block was composed of five inception modules with the following parameters for the number of filters: (192, 96, 208, 16, 48, 64), (160, 112, 224, 24, 64, 64), (128, 128, 256, 24, 64, 64), (112, 144, 288, 32, 64, 64), and (256, 160, 320, 32, 128, 128). At the end of this block, there was a 2D average pooling layer with a 3 × 3 kernel size. This block extracted a wide variety of features, which could be described as mid-level features.

The final block had a flattened layer to convert the 2D tensor into an array, followed by a dropout layer with a value of 0.4, and a dense layer with a sigmoid function to perform the binary classification task, having two neurons for that purpose. In this case, the images on the datasets were not pre-processed, i.e., the images on the datasets were the original ones.

The inception module was made of one 1 × 1 2DConv, two 3 × 3 2DConvs, two 5 × 5 2DConvs, one 3 × 3 2D Max pooling layer, and one 1 × 1 2DConv. The output of each couple of layers was then concatenated. Furthermore, for this neural network to yield relevant results in terms of learning the image patterns and generalizing for unseen cases, it needed to be regularized using the L1 regularizer with a value of 0.000001 for every trainable layer. In terms of training hyperparameters, the learning rate was set to 0.0001, with 20 training epochs and a batch size of 32. The accuracy for each epoch is presented in Figure 3a for both training and validation datasets.

The loss function determined in each epoch is also presented in a plot, Figure 3b, for both the training and validation datasets. The analysis of the values resulting from the training process, regarding the curve for the training dataset, showed that the accuracy maintained a value between 0.6 and 0.7 until the 15th epoch. Until the last epoch, the accuracy varied, reaching a final value of almost 0.8. On the other hand, on the 17th epoch, the validation set registered a global minimum for accuracy, but then reached a final value of 0.6 for the last epoch. It is possible to see on the loss graph, in Figure 3b, that for the training dataset, the loss decreased, and for the validation dataset, the loss increased, especially across the last epochs. This plot shows that some overfit was registered because the accuracy for the training dataset increased while decreasing for the validation dataset. On the loss graph, the overfit is evident by the increase in that value in the validation dataset. The resulting confusion matrix showing the values obtained from training the model for the GoogLeNet neural network is presented in Figure 4, illustrating the values obtained during the training of this model with GoogLeNet neural network.

Taking such values results in the next table, with the metrics (precision, recall, F1 score, and support) as shown for the Classes 0 and 1, as negative and positive for cancer.

In Table 3, the same metrics (precision, recall, F1 score, and support) are calculated with the macro and weighted average, as explained in the results section. The calculated metrics presented in Table 3, for Class 0, show a high recall, meaning that the number of true negatives classified was high (consequence of a low number of false positives) and a moderately high precision, showing that the number of false negatives identified was large. For Class 1, the precision was high, indicating that most of the images predicted, for that class, are true positives, and the recall was lower, indicating a high number of false negatives. The precision and recall for both classes were corroborated with each other and confirmed with the information displayed on the confusion matrix.

The F1 score of both classes considered the harmonic mean of the previous two metrics. This metric tells us that, for Class 0, the model’s performance in predicting cases from that class was reasonably effective. Instead, for Class 1, the F1 score shows that the balance between precision and recall was worse, caused by the lower recall value. Overall, this meant that this model performed better in classifying images from Class 0 than for Class 1.

Also in Table 3, the macro and weighted average values were calculated from the previous metrics. The macro average did not consider the class number when calculating that average. Instead, the weighted average considered the class weight (the number of instances).

In Figure 5a, it is possible to observe the ROC curve plotted, where the AUC (area under the curve) has a value of 0.88. The dashed blue line indicates where a random classifier would eventually fall. In the ROC curve, when the classification threshold is high, the number of false positives is low, given by the false positive rate (FPR) initial low value. The true positive rate (TPR) starts at a low value of less than 0.2, meaning that there are not so many true positive cases identified. Reaching a certain FPR value and, therefore, a higher classification threshold, leads to a higher TPR value. So, in this case, the number of true positives increases. Considering these observations, it was possible to conclude that this model could be more certain in classifying the positive instances. Until the end of the graph, the curve has a crescent tendency, plateauing at certain FPR values, suggesting that the threshold variation during a fixed TPR does not change the number of true positives. Thus, there are certain gaps during the threshold variation where no instances have been classified. In the end, the number of true positives was high, while the number of false positives is high as well, indicating that the classification threshold reached a low value. Therefore, most of the cases were classified as positives, because there was a high number of both false and true positives.

In Figure 5b, the precision–recall curve is plotted. On this curve, the classification threshold starts at a high value and decreases until the end of the graph alongside the *x*-axis. For the initial x values, those reach a threshold where the precision drops below 0.8 for a recall value close to 0.2. This means that the number of false positives increases explicitly with the precision, and, for a relatively high threshold resulting from a high number of false negatives, explicitly by the recall calculation. This is something not normal because it is expected to have only positive instances with a high predicted value, close to 1, and no negative instances. However, that is a punctual phenomenon, since the curve rises in terms of precision after that point.

After that, along with the decrease in the classification threshold, and with the decrease in the recall, given by the decrease in the number of false negatives, the precision drops again for a recall with a value of approximately 0.7, until the recall reaches the value 1. This behavior shows that the number of false positives increases explicitly with the precision, as the number of false negatives decreases, as obtained by the recall. In this case, the classification threshold has a lower value, which leads to some negative cases with a predicted value between the current classification threshold and 0.5 being classified as positives, because the instance predicted value is higher than the frontier (classification threshold) between the two classes. This is the reason why some negative instances will be false positives.

#### 3.4.2. VGG Net

In this case, the better suited VGG net for this specific classification task was VGG-11 [30]. Throughout the test, this VGG net was the one that performed better in terms of test accuracy and loss. On this neural network (NN), the parameters of the dense layers needed replacement with more convenient values for this case because the original values for those layers considered more output classes and more complex tasks. In practice, the original 4096 neurons on those layers were replaced with 64 and 128 neurons, respectively, with the last layer adjusted for the output prediction of only two neurons, i.e., the output classes. ReLU was the activation function for most of the layers while using the same padding.

Considering the five blocks preceding the last layer, the first was composed of a 2DConvs with 64 filters with a dimension of 3 and a 2D Max pooling one with 2 × 2 as the kernel size. The second had 128 filters with a kernel with the same dimensions as described before, and a 2D Max pooling layer with the same parameters as described for the previous one. The third block had two 2DConv layers, both with the same number of filters, 256, and a 2D Max pooling. The kernel dimensions were the same for every Conv2D: (3 × 3) and (2 × 2) for the pooling ones. The fourth block had two 2DConvs with 512 filters, with a max pooling after those layers. The last block before the fully connected layers was the same as the fourth block.

On the fully connected layers, there was a flattened layer that preceded the two dense layers described at the beginning of this section. After each dense layer, there was a dropout with 0.5 as its parameter value. The output layer had two neurons and a sigmoid as the activation function to perform the binary classification task. In this case, the l1 regularization technique was used with 0.00001 as its parameter, which was applied to every trainable layer. The objective of using this technique was to make the network ignore characteristics with less meaning for the classification, thus preventing overfitting [31].

The plot depicting the delta loss to the epochs was calculated through the difference between the total and L1 losses. The used images on the datasets are the originals, without any pre-processing.

Figure 6a presents the variation in the accuracy concerning the epochs during the training phase, for the training and validation datasets.

On the other hand, Figure 6b represents the delta loss variation as a function of the epochs during the training phase for the training and validation datasets. For the VGG-11 net, the training graph shows that the accuracy and loss for the training dataset followed the tendency of the same curves (accuracy and loss) for the validation dataset. For this reason, this network was generalized to classify unseen cases.

Figure 7 presents the corresponding confusion matrix, obtained using VGG-11 net with Table 4 presenting the metrics (precision, recall, F1 score, and support) for both classes, 0 and 1 (negative and positive for cancer), respectively.

For Class 0, high recall (high number of true negatives classified—a consequence of a low number of false positives) and high precision (high number of false negatives) values were found. For Class 1, the precision was high, indicating that most of the images predicted, for that class, were true positives, with some false positives. On the other hand, recall was lower, indicating a high number of false negatives. The precision and recall for both classes are corroborated with each other and are related with the information displayed in the confusion matrix. The F1 score of both classes considered the harmonic mean of the previous two metrics. This metric tells us that for Class 0, the model’s performance in predicting cases from that class was effective. Instead, for Class 1, the F1 score shows that the balance between the precision and recall was worse, caused by a lower recall. Overall, this meant that this model performed better in classifying images from Class 0 than from Class 1.

Table 4 also displays the calculated macro and weighted average values resulting from the previous metrics. The macro average did not consider the class number when calculating that average, so the precision, recall, and the F1 score calculated for that case were simply the averages. Instead, the weighted average considered the class weight (the number of instances) followed by the calculated precision, recall, and F1 score, which considered the number of instances in each class. Considering that Class 0 had more instances, its impact when calculating the weighted average was larger.

In Figure 8a, the plotted ROC curve has an AUC value of 0.82. The plotted ROC curve is similar to the previous one explained for the GoogLeNet. This has a low TPR for a low FPR, indicating a low number of true positive cases identified at a high classification threshold. When the classification threshold decreases, naturally, more true positives will be identified, reflected in the increase in the TPR value from close to 0.2 to close to 0.7. From there on, the TPR value does not increase so much anymore as the number of false positives identified increases until the end.

In Figure 8b, the precision–recall curve is plotted. Looking at the graph, there is a certain threshold where the precision drops below 0.6, for a recall value close to 0.1. This meant that the number of false positives increased, explicitly with the precision, for a relatively high threshold and for a high number of false negatives by the recall.

After that, concomitantly with the decrease in the classification threshold, and with the decrease in the recall, given by the decrease in the number of false negatives, the precision dropped, again, for a recall with a value of approximately 0.8, until the recall reached the value 1. This behavior shows that the number of false positives increased, as seen by the precision values, as the number of false negatives decreased from the recall.

#### 3.4.3. ResNet

The ResNet used in the present research was ResNet-18. The images for this case were pre-processed using the first method and resized to the requested input for this CNN, which was 224 × 224 [32].

This architecture was made of five blocks, with the first one having a 2DConv with 64 filters, 7 as the filter size, and 2 as the strides. After that layer, a batch normalization and an activation function using ReLU were used. The remaining four blocks used the residual block in pairs. So, from the first pair to the fourth, the numbers of filters were 64, 128, 256, and 512, respectively. Except for the first block, the stride values were 2 for the first residual block and 1 for the second residual block. The first block had 1 for both residual blocks.

The residual block implemented the 1 by 1 2DConv, while the remaining 2DConv had the number of filters and strides defined as before and a kernel size of 3. Then, a batch normalization layer and an activation layer (ReLU) followed. Finally, there was a 2DConv with the same parameters but with stride 1, and a subsequent batch normalization. The output of the batch normalization was summed with the shortcut, which had the 2DConv with a 1 by 1 kernel, and a batch normalization was followed by a ReLU function ending the residual block.

Figure 9 shows the confusion matrix obtained for ResNet. With these results, it becomes clear that ResNet results are all in Class 0. In this case, the training did not result in learning of patterns and after some tests, it was decided not to proceed with more experiments using with ResNet as this was not working at all.

#### 3.4.4. First Optimized Convolutional Neural Network (CNN1)

This convolutional neural network (CNN) was optimized using a random search algorithm [33,34]. To perform this optimization, the library used was scikit-learn. Before the optimization process, an interval of possible values for a defined set of hyperparameters was defined. During the optimization process, random values were tested for the hyperparameters set before. The objective of this optimization was to find a set of hyperparameters that maximized the test accuracy. These hyperparameters referred to parameters of the layers of the architecture of these CNNs (for example, the number of convolutional filters, size of those filters, optimization functions, etc.…) and to other training parameters (batch size, training epochs, etc.…).

The architecture was composed of four sets of 2DConv layers and 2D Max pooling layers. Additionally, a set of fully connected layers, which had a flattened layer, 2 dense layers, a dropout one with a value of 0.5, and an output layer were set. Some parameters were maintained for those layers. The 2DConv had a valid padding, stride with 1 and activation function with ReLU, while the pooling layer had a dimension of 2 × 2.

The set of hyperparameter intervals was the following:‘optimizer’: ‘SGD’, ‘Adam’, ‘Adagrad’, ‘RMSprop’.‘1st layer number of filters’: 4, 8.‘2nd layer number of filters’: 16, 32.‘3rd layer number of filters’: 64, 128.‘4th layer number of filters’: 512, 1024.‘1st dense layer number of neurons’: 32, 64, 128, 256, 512.‘2nd dense layer number of neurons’: 32, 64, 128, 256, 512.‘2DConv layer kernel size’: 3, 5, 7.‘epochs’: 10, 15, 20.‘batch_size’: 32.

After the optimization process, the hyperparameters that maximized the accuracy were:‘optimizer’: ‘Adam’.‘1st layer number of filters’: 8.‘2nd layer number of filters’: 32.‘3rd layer number of filters’: 128.‘4th layer number of filters’: 512.‘1st dense layer number of neurons’: 32.‘2nd dense layer number of neurons’: 64.‘2DConv layer kernel size’: 3.‘epochs’: 10.‘batch_size’: 32.

In this case, the pre-processing method used was the method previously described in the pre-processing section. In Figure 10a, it is possible to see the variation of the accuracy to the epochs, during the training phase for the training and validation datasets.

Figure 10b represents the loss variation as a function of the epochs, during the training phase for the training and validation datasets.

According to Figure 10, for CNN1, both the accuracy and loss for the training and validation datasets followed the same tendency, in terms of the numeric values registered for those metrics. During the training phase, this neural network was able to classify correctly the majority of the images in the validation dataset, which was not seen before. Consequently, this network learned the implicit patterns adequately. Figure 11 shows the corresponding confusion matrix, explicitly clarified in the results section.

As shown in Table 5, for Class 0, all metrics (precision, recall, and the F1 score) were above 0.9. This precision tells us that all instances classified as negatives were correctly identified, so they were all true negatives. Consequently, there were no false negatives. The recall value meant that some false positives were identified by the classification model. For Class 1, the precision value translated into the classification of a small number of false positives. The recall value shows that no false negatives were identified by the classification model. The precision and recall for both classes corroborate with each other and corroborate with the information displayed on the confusion matrix, supported by the existence of only three false positives. Analyzing the F1 score value, this metric shows that the classification model classified negative cases slightly better than the positives ones, because for Class 0, the value was slightly higher as compared to that found for Class 1.

Also in Table 5, both the macro and weighted averages were calculated in consequence of the previous metrics. The macro average did not consider the class number when calculating that average. Instead, the weighted average considered the class weight (the number of instances). Hence, these average values correlate with the values calculated on the table and displayed on Figure 11.

In Figure 12a, the corresponding ROC curve is plotted, showing an AUC value of 0.96, mirroring the data from Table 5. The plotted ROC curve represents a case where the TPR value increases till the unitary value, for a low FPR, less than 0.2. This indicated that most of the positive instances had a value predicted by the model very close to each other. Implicitly, this model learned to distinguish both classes with a high degree of confidence and accuracy.

In Figure 12b, the precision–recall curve is shown. In this plot, the precision–recall curve shows that the precision value had some plateaus and some drops in its value. This meant that at certain recalls and, therefore, at some classification thresholds, the number of false positives increased, but not by as many cases as the other models tested, because the precision drops were lower than the drops registered for those classification models. Besides that, the precision value was maintained higher than 0.8, indicating a good distinction between both classes. In the end, naturally, the precision value fell abruptly for a recall with value 1, since the number of false positives increased, caused by the precision value, and the number of false negatives decreased, given by the recall value.

#### 3.4.5. Second Optimized Convolutional Neural Network (CNN2)

In this case, the CNN was optimized using the Bayes optimization and the tree-structured parzen estimator (TPE) process and denoted as CNN2 [35]. As was done for the random search in CNN1, this time a new set of hyperparameters was set as the interval of each one. The set of hyperparameters consolidated not only the parameters corresponding to the numerical parameters of each layer but the number of layers (convolutional and dense), giving a broader or larger space for looking for the best set of hyperparameters that maximized accuracy.

The hyperparameters interval of values are:‘1st layer number of filters’: 4, 8, 16, 32.‘2DConv layer kernel size’: 3, 5, 7.‘Number of Conv and pooling layers (except the input layer)’: 1, 2, 3.‘2nd layer number of filters’: 8, 16, 32, 64.‘3rd layer number of filters’: 32, 64, 128.‘4th layer number of filters’: 128, 256, 512, 1024.‘Number of fully connected layers: 1, 2, 3, 4, 5.‘Number of neurons for the dense layers’: 32, 64, 128.‘dropout_rate’:0.2, 0.5;‘Learning rate’: from 1 × 10^−5^ to 1 × 10^−1^.

After the optimization the hyperparameters determined to maximize the accuracy were:‘1st layer number of filters’: 8.‘2DConv layer kernel size’: 3.‘Number of Conv and pooling layers (except the input layer)’: 2.‘2nd layer number of filters’: 8.‘3rd layer number of filters’: 64.‘Number of fully connected layers: 3.‘Number of neurons for the first dense layers’: 32.‘Number of neurons for the second dense layers’: 64.‘Number of neurons for the third dense layers’: 64.‘dropout_rate’:0.5.‘Learning rate’: 0.00233.

The library used to implement this optimization process was Optuna. The images were pre-processed using the second method described in the respective section. In Figure 13, it is possible to see the variation in the accuracy and the loss across the epochs, during the training phase for the training dataset.

During the CNN2 training phase, the accuracy and loss results for the training dataset indicated that the learning process was successful, as shown by the increase in the accuracy and the corresponding decrease in the loss till the last epoch. One more thing that supported this claim was that the test results for the accuracy and loss corresponded to the accuracy and loss registered on the plot in the last epoch. This meant that the network could generalize to unseen cases, based on the training dataset images. Figure 14 shows the corresponding confusion matrix arising from these results, explicitly displaying the results of the classification.

As evidenced in Table 6, for Class 0, all metrics (precision, recall, and the F1 score) were above 0.9. This precision tells us that all instances classified as negatives were correctly identified, so they were all true negatives and, consequently, there were no false negatives. The recall value meant that some false positives were identified by the classification model. For Class 1, the precision value translated into the classification of a certain number of false positives. The recall value showed that no false negatives were identified by the classification model. The precision and recall for both classes corroborated with each other and with the information displayed in the confusion matrix (Figure 14), given by the existence of only two false positives. Analyzing the F1 score, this metric shows that the classification model classified negative cases slightly better than the positives, given that for Class 0 the value was slightly higher, compared to that found for Class 1.

Also in Table 6, both the macro and weighted averages were calculated in consequence of the previous metrics, with all values close to unity.

In Figure 15a, the plotted ROC curve reached an AUC value of 0.99. Just like the ROC curve plotted for CNN2, this case shows an even better distinction between the two classes, because of the TPR value sharp rise at low FPR values.

In Figure 15b, the precision–recall curve shows an almost horizontal curve between 0 and 0.8, for the recall. This means that no false positive cases have been classified when the recall increased along with the classification threshold decrease. For recall values above 0.8, some false positive cases were identified (Figure 14). This curve profile suggests that the two classes were well distinguished by the classification model.

## 4. Discussion

The results of the application of the selected model algorithms to this classification problem are summarized in Table 7 and Table 8.

From the analysis of Table 7, GoogLeNet achieved the highest precision for Class 1 and the highest recall for Class 0, overall. Both CNN1 and CNN2 algorithms yielded the highest values for precision and recall for Classes 0 and 1, respectively, expressed as the inexistence of false negative cases for both models. On the other hand, GoogLeNet only had one false positive case identified, which was the reason for the respective identified metrics. Considering that the F1 score was calculated based on the harmonic mean of the two optimized CNN models, they evidenced the highest values due to their high precision and recall values. This stressed the balanced profile of these classification models as the best performing ones.

From the results shown in Table 8, it is possible to confirm that from the three standard architectures—GoogLeNet, VGG, and ResNet—the former was the most suitable network for this binary classification task in terms of accuracy and loss of the evaluation of the test dataset. Considering the 5-fold cross-validation, in this case, the stratified cross-validation was used to maintain the proportion of the classes. Comparatively speaking, the average accuracy and loss of the GoogLeNet during the cross-validation was better than any results obtained for all other models, exhibiting the robustness of this network in classifying different test datasets based on different training datasets. Looking at the accuracy and loss variation with the epochs during the training phase, it is evident that some overfit was present at the last epoch because the accuracy of the validation dataset decreased, and the loss of the validation dataset increased. When comparing with the same metrics for the training dataset, this tells us that the network stopped learning the patterns of the images, meaning that the accuracy and loss of the validation dataset deteriorated. However, the impact on results for the test dataset evaluation was not significant, shown by the accuracy of 84% and the loss of 0.49 which correlated or were similar to values in the last epoch of accuracy and loss plots, showing that this model generalized well for unseen cases.

Globally speaking, all tested models and architectures were capable of generalizing for unseen images. This means that the training phase was successfully accomplished and that the model could learn the implicit patterns of the spectrographic image. In terms of the classification of the test dataset, both the accuracy and the loss of the CNN2 model showed that those values were the highest and the lowest obtained, respectively. Besides that, the 5-fold stratified cross-validation indicates that the average accuracy was lower and the loss higher, evidenced by the lack of capability to maintain the performance shown on the test dataset.

Regarding the AUC, it was possible to see that the CNN resulting from the random search process of optimization had the highest AUC. Nonetheless, with the exception of the ResNet algorithm, all models showed high AUC, demonstrating the capability to distinguish between the two classes.

Despite ResNet being a promising CNN, it was not possible to reach a point where it could be possible to make the network learn the patterns to make the classification task with minimal distinction between the two labels. Hence, the accuracy value showed that all images were classified in the majority class, not using the patterns to effectively distinguish both classes.

Looking at both the ROC and precision–recall curves, it was possible to see that globally every network had a regular shape close to the ideal shape for both curves. However, VGG-11 had a less ideal shape for both curves, showing the fragilities of this model in some classification thresholds where the number of false positives and negatives could increase. The precision–recall curve and the ROC curve both showed the existence of plateaus after decreases, identical to a saw shape. The first graph indicates the fluctuations in false positives because the threshold was becoming lower with the increase in the recall’s value, and the number of positives identified was higher, although some of those were false positives. For the ROC curve, the vertical variation in the position of the plateaus indicated the variability in the true positive cases.

## 5. Conclusions

The research presented in this document aimed to analyze the potential use of several algorithms to evaluate the chances for both Classes: 0 for negative and 1 for lung cancer. For the study, the most widely utilized architectures in the literature were screened and regularized to obtain a good classification performance, which indeed was achieved. During the training cycles, different optimization methods were used to determine the best set of hyperparameters to maximize the accuracy of each one of them. Considering all the models, obtained with different architectures, the best one resulted from the Bayes optimization, having 96% accuracy and 0.23 loss from the test dataset evaluation. In terms of the evaluation of the model using 5-fold cross-validation, the best performing model was GoogLeNet with approximately 84% average accuracy and 0.47 average loss. By the same token, the CNN2 model also presented an almost identical performance in the cross-validation, hitting 81% average accuracy and 0.56 average loss. These results demonstrate the value of the agnostic screening approach, proposed in the present study, as an alternative to seeking for specific compounds. The results achieved so far with the proposed algorithms are quite promising and the next step is to evolve towards the distinction between different lung cancer stages, I to IV. In the future, since, in most cases, no VOC records exist for each patient, the usage of AI to correlate physiological parameters with VOC records may be able to promote the personalized classification of risk for each patient.

## Figures and Tables

**Figure 1 cancers-16-02200-f001:**
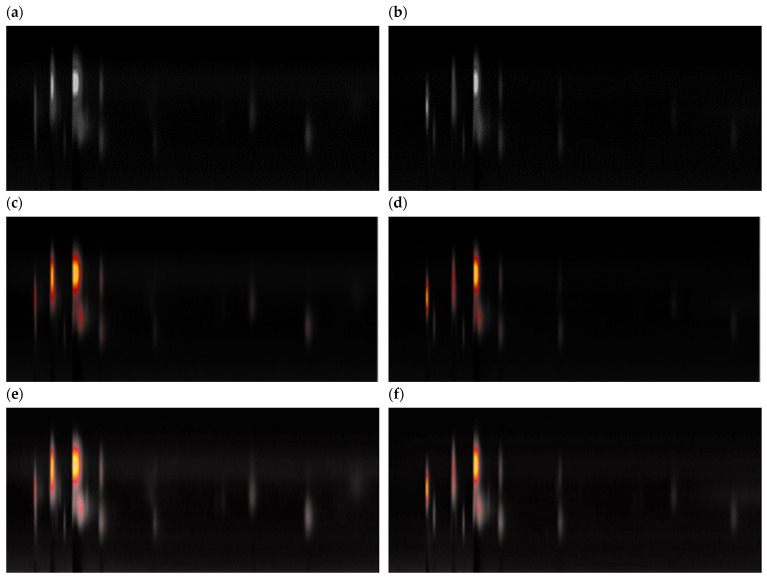
Image of exhaled breath profiles (chromatograms) obtained by GC-FAIMS analysis from a lung cancer patient (**a**,**c**,**e**) and from a healthy control individual (**b**,**d**,**f**). Images were represented as grayscale (**a**,**b**), RGB (**c**,**d**), and RGB-equalized by contrast-limited adaptative histogram equalization (CLAHE) (**e**,**f**) [26].

**Figure 2 cancers-16-02200-f002:**
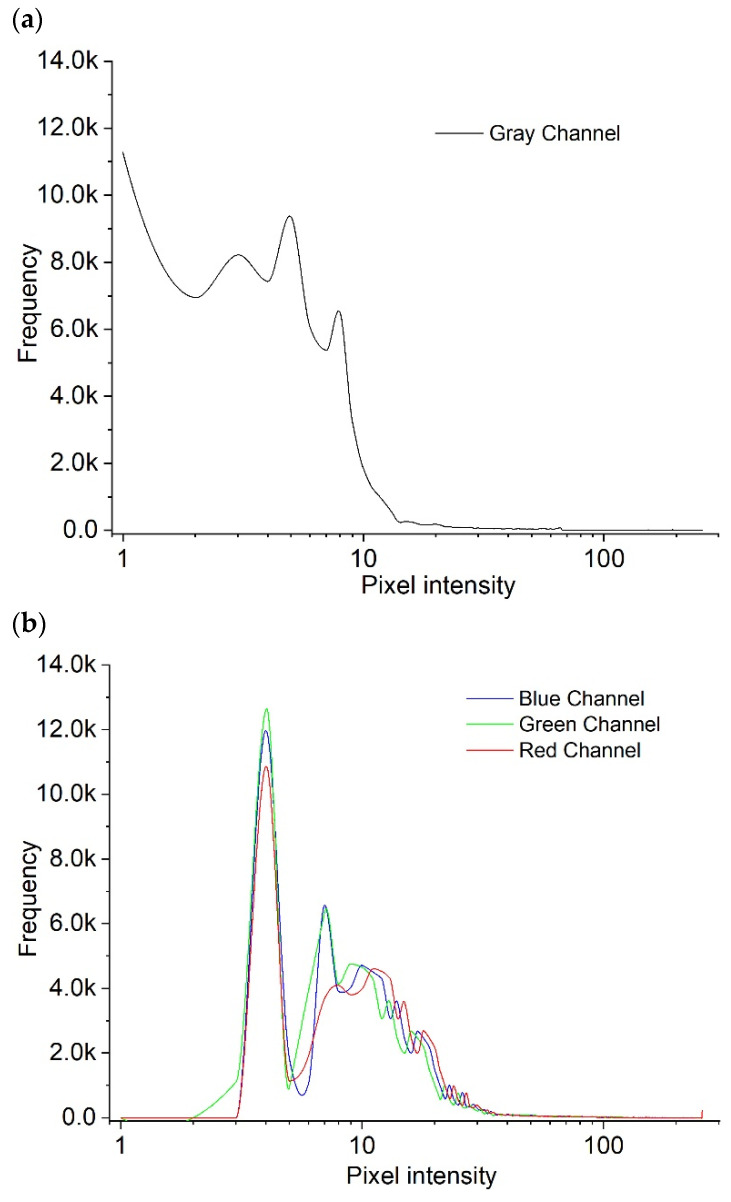
Equalized histograms of a grayscale image (**a**) and of the corresponding original image (**b**) using CLAHE.

**Figure 3 cancers-16-02200-f003:**
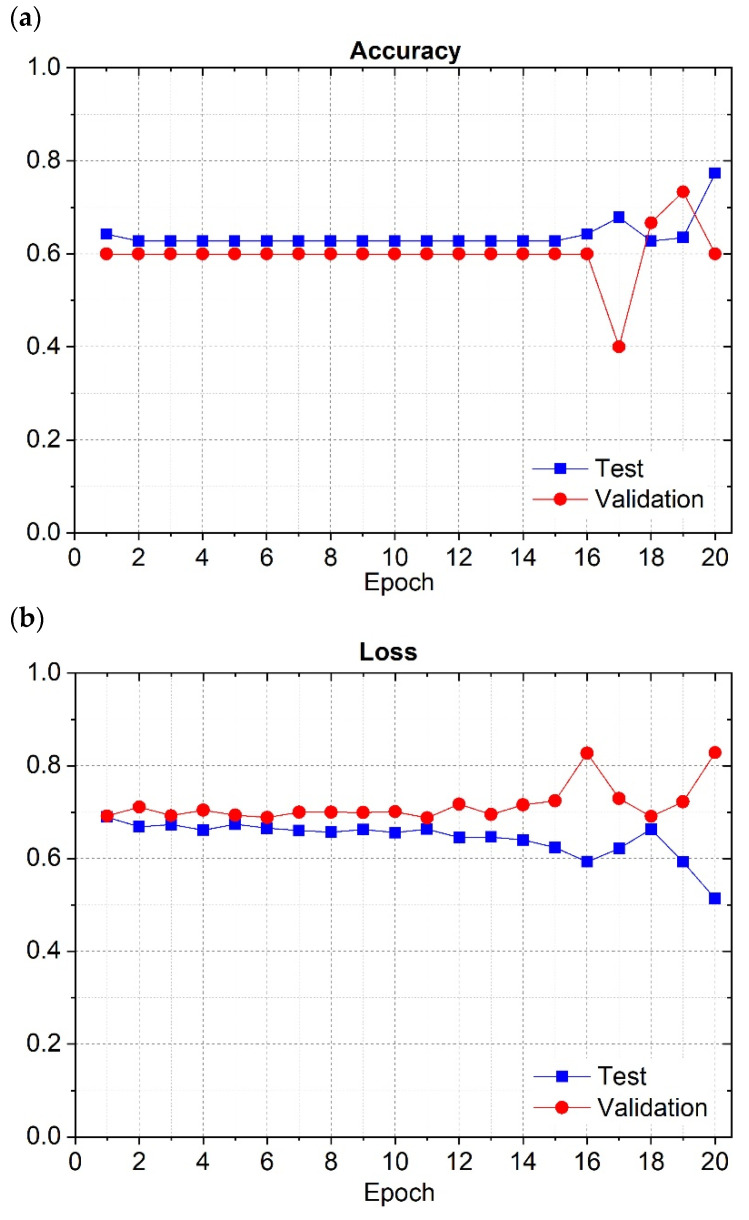
GoogLeNet’s accuracy (**a**) and loss (**b**) during the training phase for the training and validation datasets.

**Figure 4 cancers-16-02200-f004:**
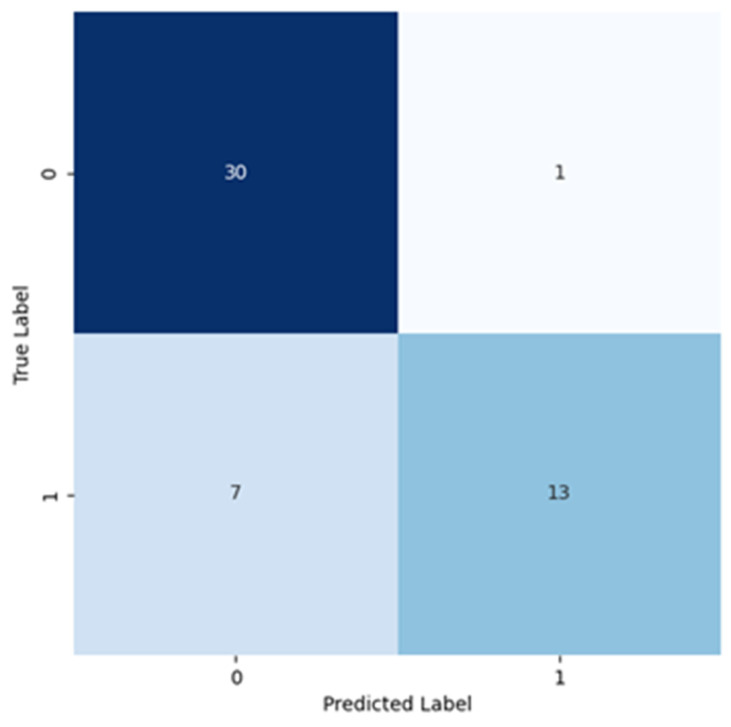
Confusion matrix resulting from the test of the model trained using GoogLeNet.

**Figure 5 cancers-16-02200-f005:**
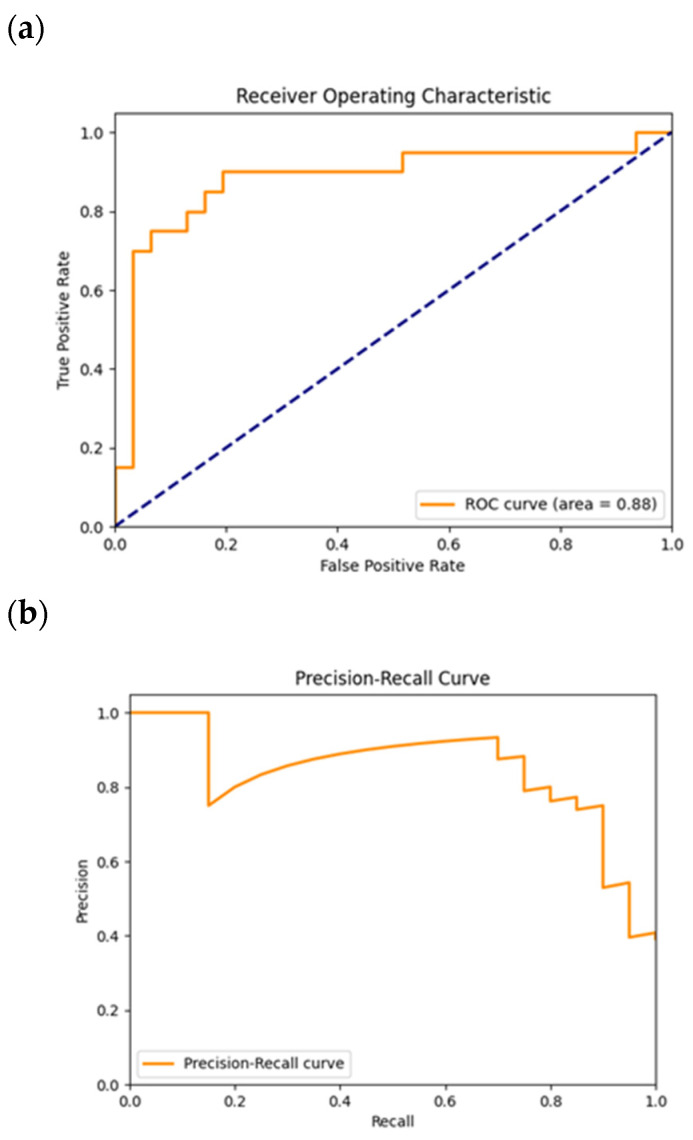
ROC (**a**) and precision–recall (**b**) plots obtained from the test of the model trained for the GoogLeNet. The dashed diagonal line in (**a**) corresponds to the random guess curve, representing the least accurate hypothesis.

**Figure 6 cancers-16-02200-f006:**
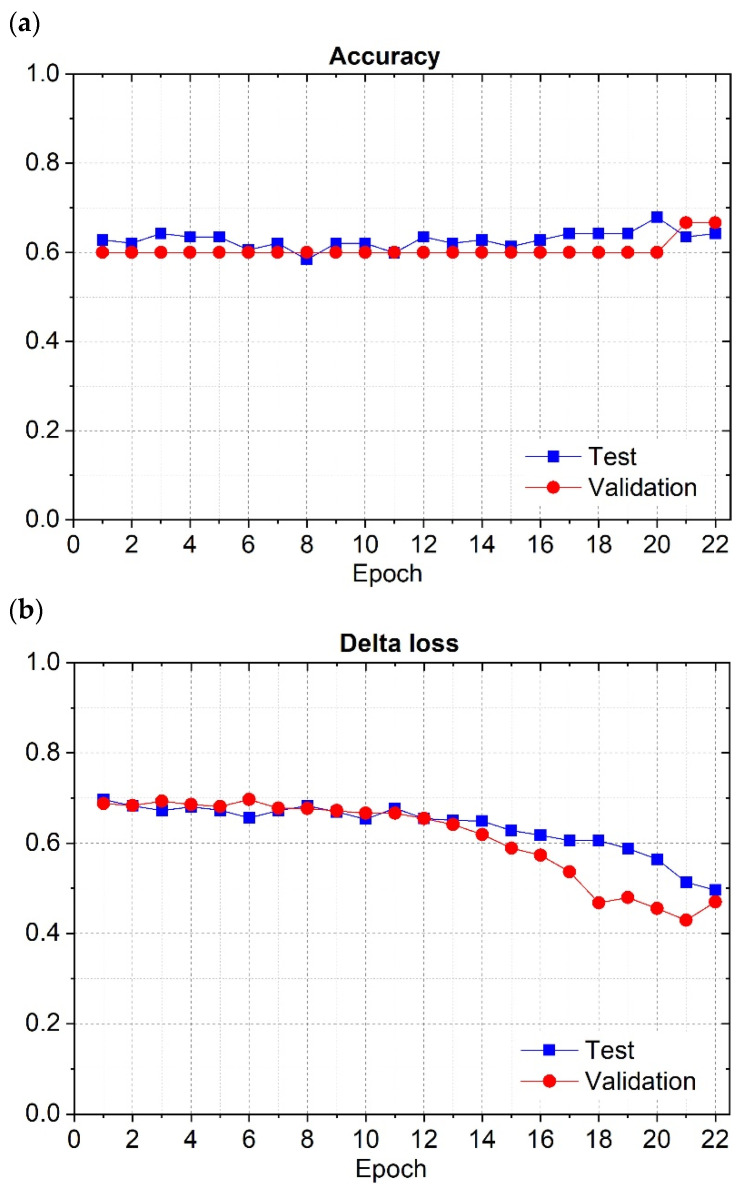
VGG-11′s accuracy (**a**) and delta loss (**b**) during the training phase for the training and validation datasets.

**Figure 7 cancers-16-02200-f007:**
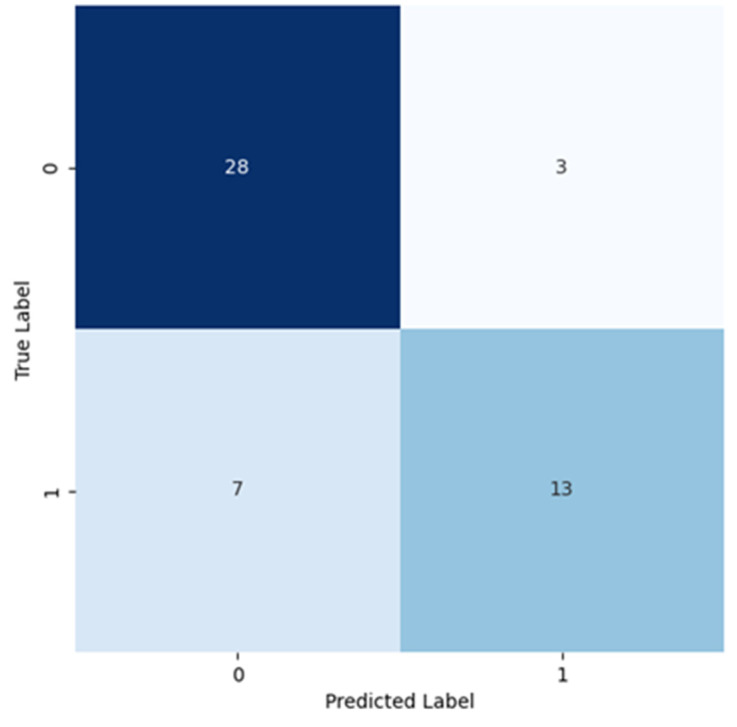
Confusion matrix resulting from the test of the model trained using VGG-11.

**Figure 8 cancers-16-02200-f008:**
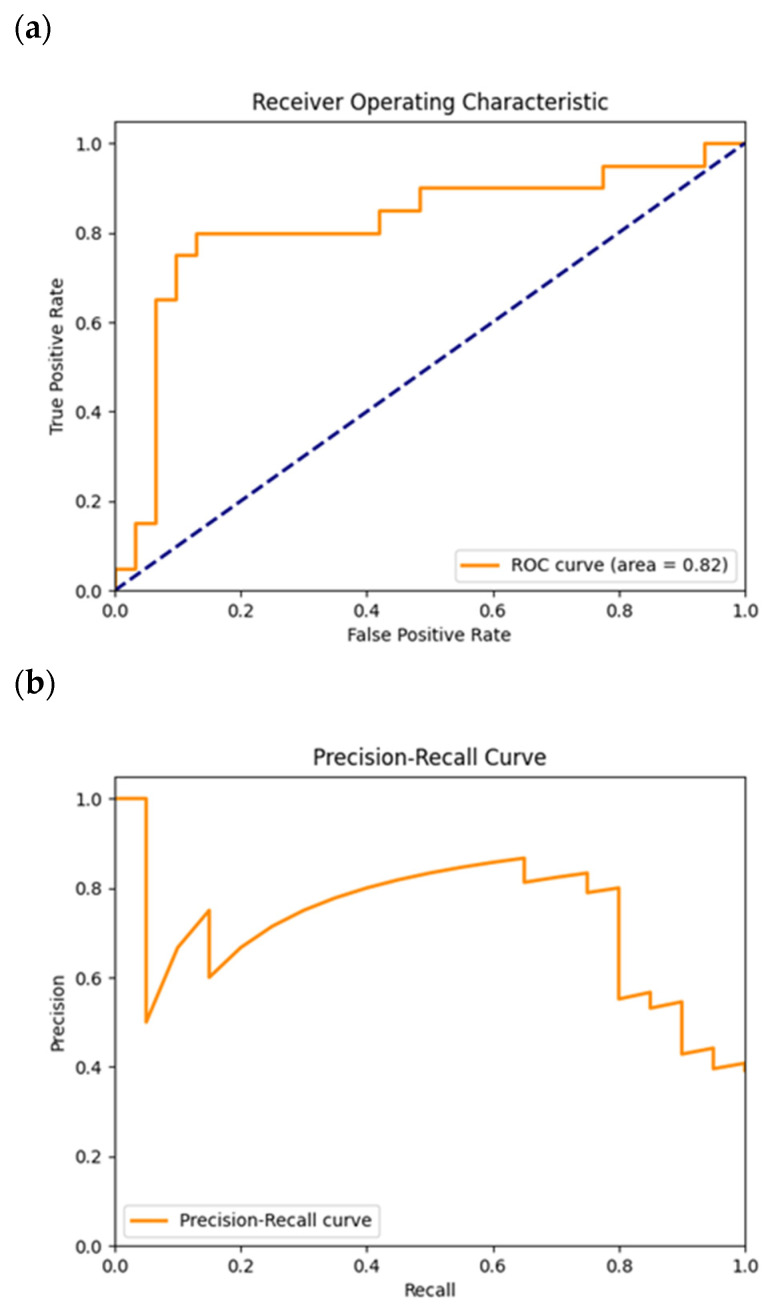
ROC (**a**) and precision–recall (**b**) plots obtained from the test of the model trained for the VGG-11 net. The dashed diagonal line in (**a**) corresponds to the random guess curve, representing the least accurate hypothesis.

**Figure 9 cancers-16-02200-f009:**
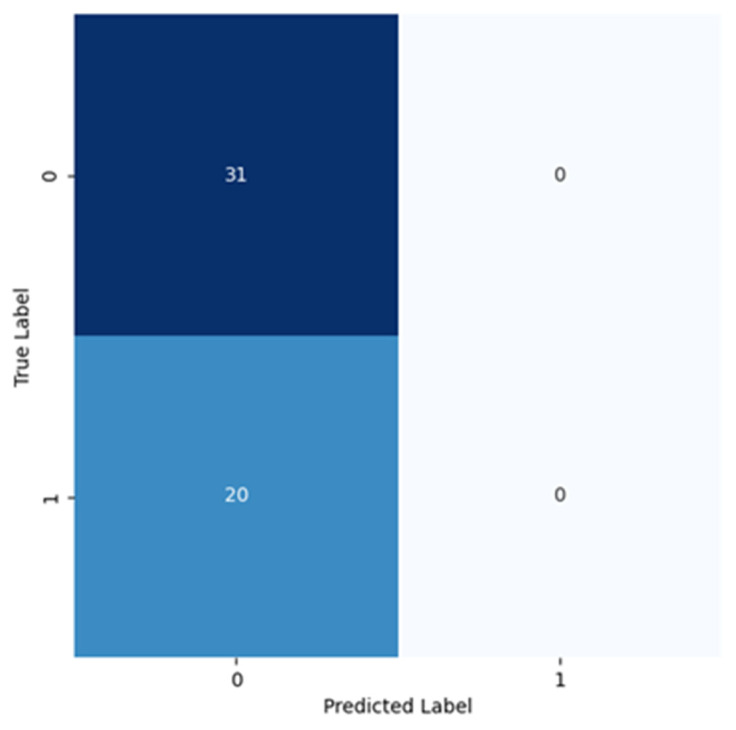
Confusion matrix resulting from the test of the model trained using ResNet-18.

**Figure 10 cancers-16-02200-f010:**
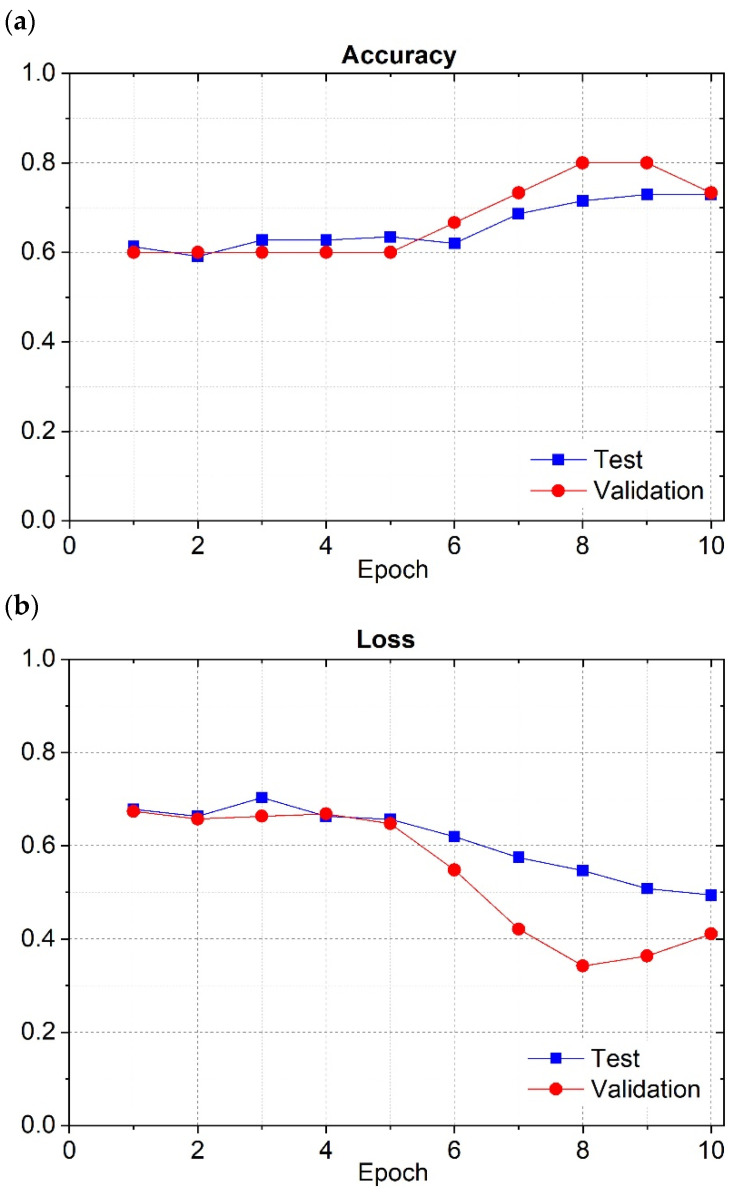
First-optimized CNN’s (CNN1) accuracy (**a**) and loss (**b**) during the training phase for the training and validation datasets.

**Figure 11 cancers-16-02200-f011:**
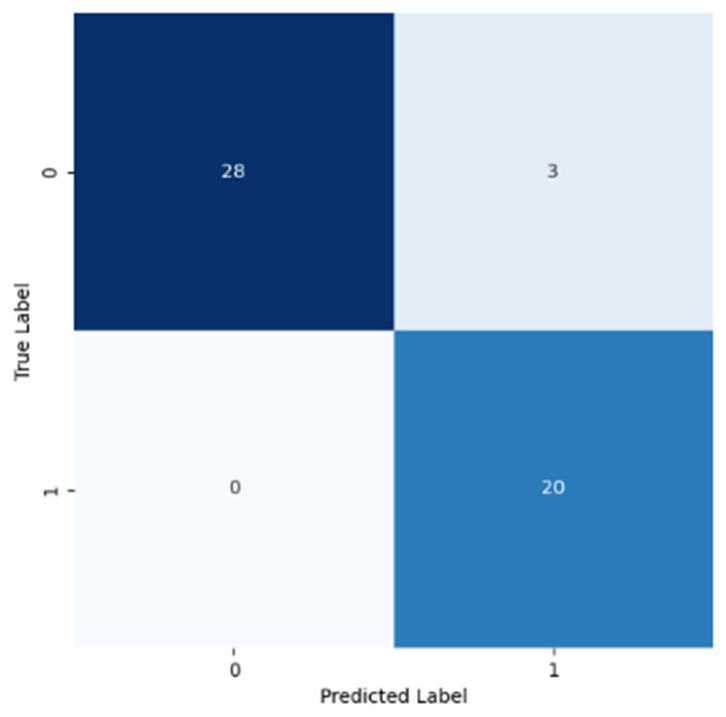
Confusion matrix resulting from the test of the model trained using the first optimized CNN (CNN1).

**Figure 12 cancers-16-02200-f012:**
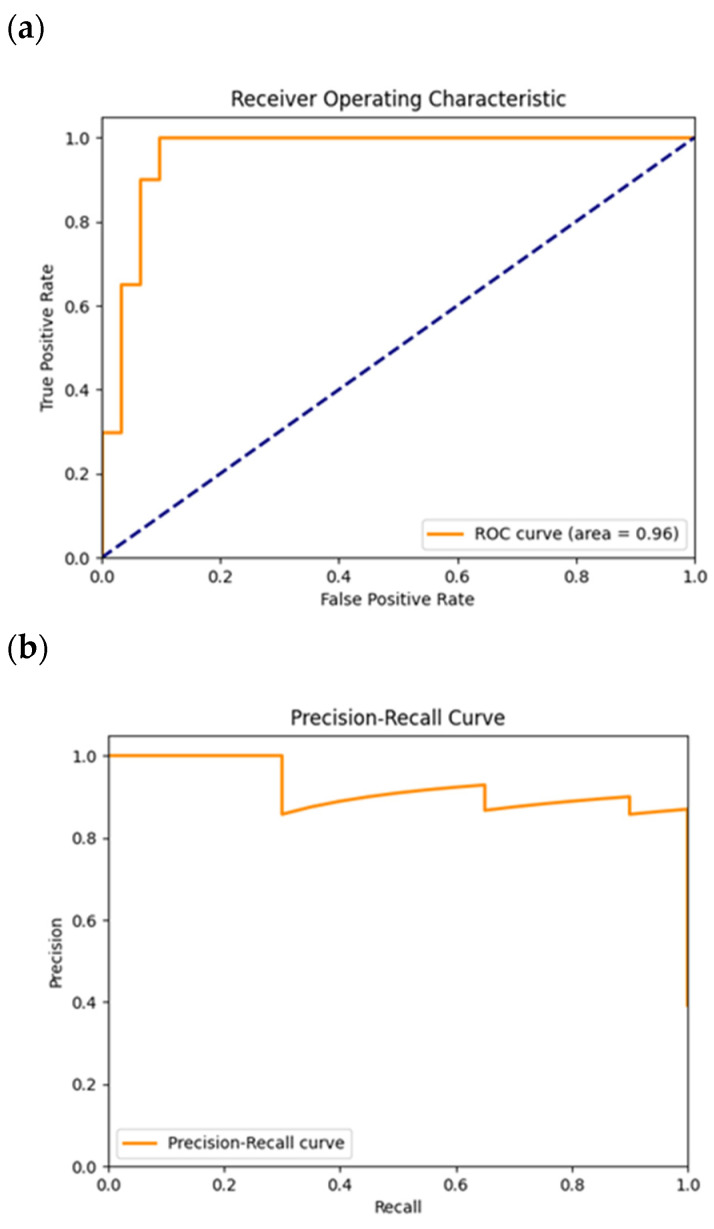
ROC (**a**) and precision–recall (**b**) plots obtained from the test of the model trained for the CNN1. The dashed diagonal line in (**a**) corresponds to the random guess curve, representing the least accurate hypothesis.

**Figure 13 cancers-16-02200-f013:**
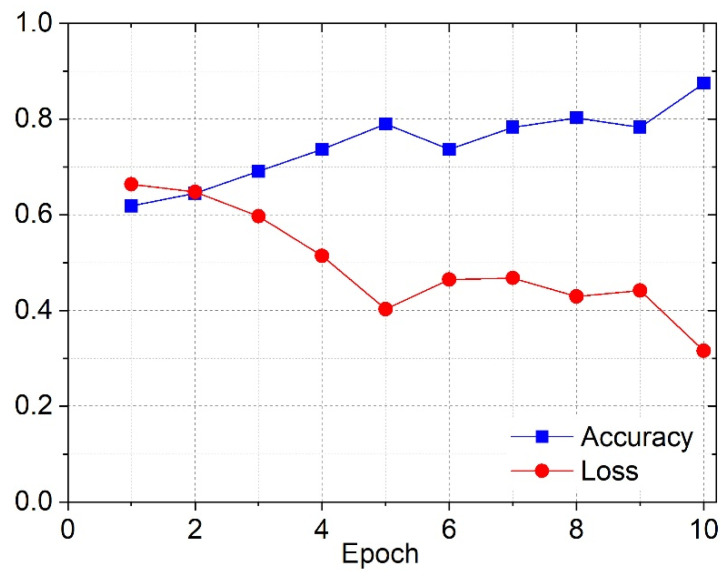
Accuracy and loss during the training phase for training dataset using the second-optimized CNN (CNN2).

**Figure 14 cancers-16-02200-f014:**
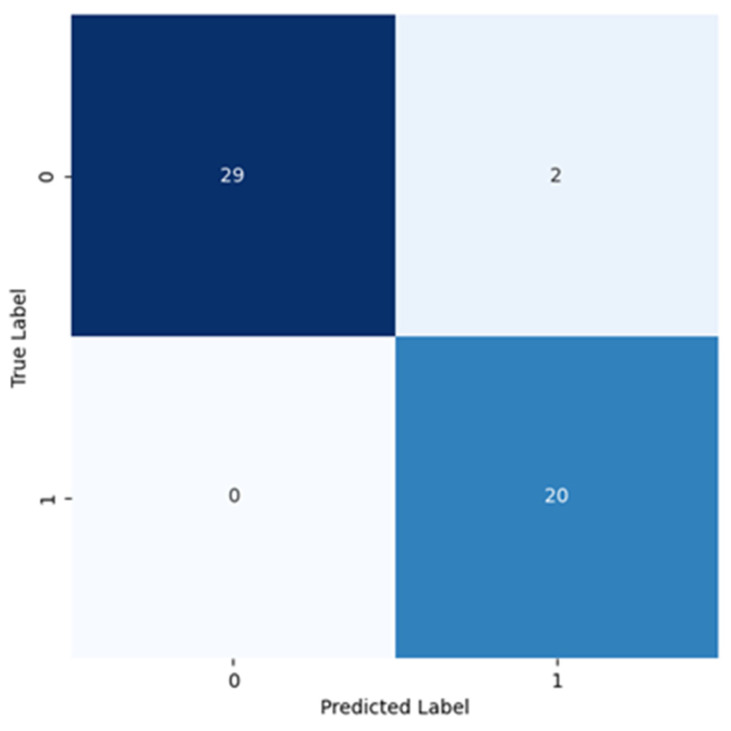
Confusion matrix resulting from the test of the model trained using the second optimized CNN (CNN2).

**Figure 15 cancers-16-02200-f015:**
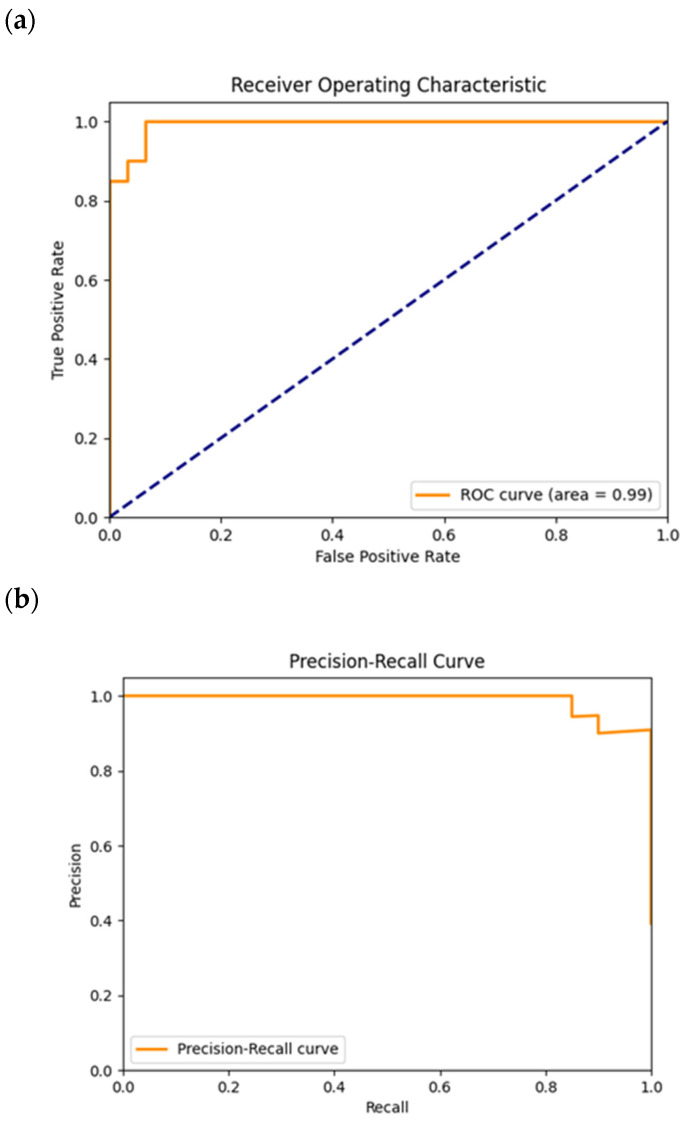
ROC (**a**) and precision–recall (**b**) plots obtained from the test of the model trained for the CNN2. The dashed diagonal line in (**a**) corresponds to the random guess curve, representing the least accurate hypothesis.

**Table 1 cancers-16-02200-t001:** Characterization of the recruited volunteers across both lung cancer and healthy control groups of the study.

Parameter		Lung Cancer Patients	Healthy Controls
*n* (%)		77 (37.9)	126 (62.1)
Age (years)			
	Median	66	40
	Range	41–86	20–78
Sex, *n* (%)			
	Female	49 (63.6)	96 (76.2)
	Male	28 (36.4)	30 (23.8)
Smoking status, *n* (%)			
	Current smoker	24 (31.2)	39 (30.7)
	Ex-smoker	29 (37.6)	17 (13.4)
	Never-smoker	24 (31.2)	71 (55.9)

**Table 2 cancers-16-02200-t002:** Dataset description.

Dataset	Class 0	Class 1	Total
Training	86	51	137 (68%)
Validation	9	6	15 (7%)
Test	31	20	51 (25%)

**Table 3 cancers-16-02200-t003:** Precision, recall, F1 score, and support obtained from the model training test for both classes and the corresponding averages of the final model using GoogLeNet.

Classes	Precision	Recall	F1 Score	Support
**0**	0.81	0.97	0.88	31
**1**	0.93	0.65	0.76	20
**Average**				
**Macro**	0.87	0.81	0.82	51
**Weighed**	0.86	0.84	0.84	51

**Table 4 cancers-16-02200-t004:** Precision, recall, F1 score, and support obtained from the model training test for both classes and the corresponding averages of the final model using VGG-11 net.

Classes	Precision	Recall	F1 Score	Support
**0**	0.80	0.90	0.85	31
**1**	0.93	0.65	0.72	20
**Average**				
**Macro**	0.81	0.78	0.79	51
**Weighed**	0.80	0.80	0.80	51

**Table 5 cancers-16-02200-t005:** Precision, recall, F1 score, and support obtained from the model training test for both classes and the corresponding averages of the final model using CNN1.

Classes	Precision	Recall	F1 Score	Support
**0**	1.00	0.90	0.95	31
**1**	0.87	1.00	0.93	20
**Average**				
**Macro**	0.93	0.95	0.94	51
**Weighed**	0.95	0.94	0.94	51

**Table 6 cancers-16-02200-t006:** Precision, recall, F1 score, and support obtained from the model training test for both classes and the corresponding averages of the final model using CNN2.

Classes	Precision	Recall	F1 Score	Support
**0**	1.00	0.94	0.97	31
**1**	0.91	1.00	0.95	20
**Average**				
**Macro**	0.95	0.97	0.96	51
**Weighed**	0.96	0.96	0.96	51

**Table 7 cancers-16-02200-t007:** Compilation of the performance metrics from the test dataset evaluation using across all models screened.

Model	Classes	Precision	Recall	F1 Score
**GoogLeNet**	0	0.81	0.97	0.88
	1	0.93	0.65	0.76
**VGG-11**	0	0.80	0.90	0.85
	1	0.81	0.65	0.72
**CNN1**	0	1.00	0.90	0.95
	1	0.87	1.00	0.93
**CNN2**	0	1.00	0.94	0.97
	1	0.91	1.00	0.95

**Table 8 cancers-16-02200-t008:** Compilation of the accuracy, loss, AUC, and 5-fold stratified cross-validation for across all models screened.

Model	Accuracy	Loss	AUC	5-Fold Cross Validation
**GoogLeNet**	0.8431	0.4941	0.88	Average accuracy: 0.8373
				Average loss: ^[1]^ 0.4669
**VGG**	0.8039	0.5954	0.82	Average accuracy: 0.7834
				Average loss: ^[1]^ 1.0000
**ResNet**	0.6078	0.7291	-	-
**CNN1**	0.9412	0.4992	0.99	Average accuracy: 0.7929
				Average loss: 0.6148
**CNN2**	0.9607	0.2266	0.96	Average accuracy: 0.8130
				Average loss: 0.5613

^[1]^ The average loss obtained in these cases refers to the total loss.

## Data Availability

The data that made it possible to conduct this study are available from the corresponding author on reasonable request and once approved by the Data Protection Officer from author’s affiliation institution.

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
