# Peer review of "AI Applied to Volatile Organic Compound (VOC) Profiles from Exhaled Breath Air for Early Detection of Lung Cancer"

_cancers, 2024, doi:10.3390/cancers16122200_

Round 1

Reviewer 1 Report

Comments and Suggestions for Authors

The manuscript discusses an approach using Volatile organic compounds (VOCs) for detecting lung cancer. This approach also uses AI and machine learning tools in conjunction with VOCs to improve the accuracy and detection efficiency of lung cancer. This approach used exhaled breath air from patients and used that data to detect lung cancer. This study uses an agnostic approach for maximizing the detection efficiency over the identification of specific compounds. This was performed to focus on the compositional profile and to evaluate the differences across different group of patients.

There are some notable strengths of the manuscripts,

1.     The manuscript effectively outlines the approach comparing different Deep learning models for the task of lung cancer detection using the images from the VOC analysis.

2.     The approach used in the manuscript makes it more likely for people to use this screening method because it is comfortable and non-invasive, like a simple breath test, and the results show that it is reliable for detecting lung cancer at early stages.

Few Questions / Comments on the manuscript are as follows,

1.     Section 3.3, Dataset Structure, the authors mention about trimming the non-relevant information. The authors did not list the criteria used for trimming the current dataset. This information would be useful to understand the data pre-processing done before any analysis.

2.     Section 3.3, Dataset Structure, the line, ‘mostly by trimming the non-relevant information, thus resulting in two distinct datasets. Each dataset was used to train, validate, and test the model.’ was not clear. Does the author mean each dataset from the patient was used in training validation and testing set? If so, this would create a data leakage in the training and test. This information regarding the dataset was not clearly presented in this section.

3.     Section 3.3, Dataset Structure, this line needs to be edited grammatically, ‘Those labels matched each sample on the image datasets with the corresponding lung cancer situation that, in the present study would be classified in a binary mode 1 or 0 for cancer or healthy, respectively.

4.     Section 3.3, Dataset Structure, this line needs to be edited grammatically, ‘The objective of this method was to make some hidden (not so bright) patterns of the image more evident.

5.     Section 3.4.1, GoogLeNet, this line needs to be edited grammatically, ‘The resulting confusion matrix is presented next with the values obtained in the training of the model trained for the GoogLeNet neural network’.

6.     Section 3.4, Experimental methodology, Loss Function used in the experiment was not mentioned in the manuscript. This information would be useful for reproducibility.

7.     Section 3.4.4, this line seems incomplete and requires additional text, ‘This Convolutional Neural Network (CNN) was optimized using a random search.

8.     Since, both the CNNs were optimized using scikit learn library, the authors did not mention the dataset used for optimizing the CNN models.

9.     Section 3.4, Experimental Methodology, the line needs to be edited grammatically, ‘For each tested model a table with the accuracy, loss, AUC (Area Under the Curve) and cross-validation was created was created wrapping up all results’.

10.  The methodology section mentioned that a cross-validation was conducted on all the models. But the result table doesn’t display a standard deviation value for result table, specifically for the metrics such as Precision, recall, and F1-score

Comments on the Quality of English Language

The quality of English used in the manuscript was ok

Reviewer 2 Report

Comments and Suggestions for Authors

This mansucript presented several algorithms for evaluating the accuracy of diagnosing lung cancer via breath analysis. The overall stragtegy is accepatble, while more details should be additionally stated.

1. The most challenging issue for breath alaysis is the individual difference, while the impact of these factors have not been discussed in the mansucript. It is better to add part of statement to show how to figure out this issue by AI technology in future;

2. It is better to compare the time that consumed for each algorithms when processing data, since the future application also needs to consider the efficiency. 

Comments on the Quality of English Language

Good

Reviewer 3 Report

Comments and Suggestions for Authors

The present study is using artificial intelligence and machine learning algorithms to enhance classification accuracy and efficiency in detecting lung cancer through VOC analysis collected from exhaled breath air using ReCIVA breath sampler for sample’s collection and GC-FAIMS for sample’s analysis. Different models and architectures were used, all demonstrating the capability to distinguish between the two classes investigated.

Overall, the article is well written and organized, the results are well presented and can support the conclusions. Please see my comments and suggestions below.

I will shortly point out in the introduction some analytical techniques currently used for detection of VOCs including those VOCs related to lung cancer

Does the approved of the ethics committee has a number? Please mention the number in the article.

The authors wrote in the article “Patients who fulfilled the eligibility criteria….”. Please write in the article what were the eligibility criteria.

Please describe how was the blanks’ collection procedure.

In how long time the patients succeeded to donate 2L of sample?

The number of healthy volunteers is considerably higher compare with the LC subjects, while the age difference of groups if also big. Why the authors did not collect the same number of samples form LC patients and volunteers and they did not try to manage better the age difference?

Honestly, I cannot see a relevant difference between the chromatograms of LC group and healthy control group in Figure 1

If possible, please increase the quality of Figure 2, especially part a; Similar comment for Figure 3, 4, 7, 9, 11, 14

Round 2

Reviewer 3 Report

Comments and Suggestions for Authors

The authors answered my questions and the manuscript was improved. I recommend to be publish in the present form.